# Comparable Outcomes in Early Hepatocellular Carcinomas Treated with Trans-Arterial Chemoembolization and Radiofrequency Ablation

**DOI:** 10.3390/biomedicines10102361

**Published:** 2022-09-22

**Authors:** Benjamin Wei Rong Tay, Daniel Q. Huang, Muthiah Mark, Neo Wee Thong, Lee Guan Huei, Lim Seng Gee, Low How Cheng, Lee Yin Mei, Prem Thurairajah, Lim Jia Chen, Cheng Han Ng, Wen Hui Lim, Darren Jun Hao Tan, Da Costa Maureen, Kow Wei Chieh Alfred, Iyer Shridar Ganpathi, Tan Poh Seng, Dan Yock Young

**Affiliations:** 1Division of Gastroenterology and Hepatology, National University Health System, Singapore 119228, Singapore; 2Department of Medicine, Yong Loo Yin School of Medicine, National University of Singapore, Singapore 119077, Singapore; 3Department of Diagnostic Imaging, National University Health System, Singapore 119228, Singapore; 4Division of Hepatobiliary Surgery, National University Health System, Singapore 119228, Singapore

**Keywords:** transarterial chemoembolization, radiofrequency ablation, hepatocellular carcinoma

## Abstract

The guidelines recommend radiofrequency ablation (RFA) for early hepatocellular carcinomas that are less than 3 cm and trans-arterial chemoembolization (TACE) for intermediate-stage tumors. Real-world patient and tumor factors commonly limit strict adherence to the guidelines. We aimed to compare the clinical outcomes for TACE and RFA in early HCC. All consecutive patients from 2010 to 2014 that were treated with locoregional therapy at our institution were enrolled. The decision for TACE or RFA was based on tumor location, stage and technical accessibility for ablation. A subgroup analysis was performed for patients with tumors less than 3 cm. A total of 168 patients underwent TACE while 56 patients underwent RFA. Patients treated with TACE and RFA had 1- and 5-year survival rates of 84.7% and 39.8% versus 91.5% and 51.5%, respectively (*p* = 0.28). In tumors less than 3 cm, there was no significant difference in overall survival (*p* = 0.69), time to progression (*p* = 0.55), or number of treatment sessions required (*p* = 0.12). Radiofrequency ablation had a significantly higher chance of a complete response (*p* = 0.004). In conclusion, TACE may be selectively considered for early-stage hepatocellular carcinoma in patients unsuitable for other modalities.

## 1. Introduction

Hepatocellular carcinoma (HCC) is the third leading cause of cancer-related death worldwide [1,2]. The optimal treatment for HCC is oncologic resection or liver transplant [3,4,5,6]. Unfortunately, this is often limited by poor liver function precluding a safe resection and a shortage of donor livers. In situations when a resection or transplant is not possible and the disease is confined to the liver, locoregional therapies for HCC are recommended, including radiofrequency ablation (RFA) and transarterial chemoembolization (TACE) [7,8].

RFA delivers a high-frequency alternating current via a catheter tip, producing thermal-energy-induced area necrosis to tissue within a 1.5 cm radius. RFA is best used in tumors with a diameter of less than 3 cm and can provide sustained recurrence-free survival in these patients. In major society guidelines, RFA is recommended for small HCCs less than 3 cm in size, especially if resection is not feasible [7,8,9]. RFA is generally avoided in tumors that are near the dome of the diaphragm or next to bowel lumen, for fear of diaphragmatic injury or bowel perforation [10,11].

TACE delivers chemotherapy-infused particles via the hepatic artery, inducing ischemia in the tumor and delivering the local chemotherapeutic agent. Previously, a Taiwanese study showed that TACE and RFA resulted in similar overall survival for HCC patients within the Milan criteria, although RFA still showed significantly better survival in small tumors with a total tumor volume < 11 cm³ [12]. However, Kim et al. compared TACE and RFA for small tumors less than 2 cm and showed a similar overall survival in both groups, and another study comparing single small tumors less than 3 cm showed a similar tumor response and recurrence [13,14]. The current practice guidelines recommend ablation in HCCs less than 3 cm. Considering the conflicting literature with regards to the use of TACE in small tumors, and the current clinical practice guidelines recommending ablation in HCCs less than 3 cm, we aimed to compare the outcomes of TACE and RFA in patients with HCCs less than 3 cm. 

## 2. Materials and Methods

### 2.1. Patients

This was a single-center retrospective cohort study. We enrolled all consecutive patients between 1 January 2010 to 31 December 2014 treated with either TACE or RFA as the first line monotherapy for Barcelona Clinic Liver Cancer (BCLC) stage 0, A or B HCC. Patients with evidence of vascular invasion or metastatic disease were excluded. This study was approved by the Investigation and Ethics Committee of the National University Hospital (Singapore), according to the standards of the Declaration of Helsinki. Written informed consent was obtained from each patient before treatment. 

### 2.2. Selection of Primary Treatment Modality

The diagnosis of HCC was in accordance with major society guidelines [7,8,15]. The selection of the locoregional modality was made by a multidisciplinary hepatobiliary tumor board consisting of hepatobiliary surgeons, hepatologists, interventional radiologists, medical/radiation oncologists and pathologists. Locoregional therapy was offered to patients who were not surgical candidates due to a combination of poor liver function, poor functional status and/or multiple comorbidities. The decision between RFA and TACE depended on factors such as tumor location, size, number of nodules and technical feasibility. 

### 2.3. Trans-Arterial Chemoembolization

Dynamic multiphasic cross-sectional imaging of the liver via magnetic resonance imaging (MRI) or computed tomography (CT) was performed prior to the procedure to guide the approach to the tumor. Catheterization of the hepatic artery and identification of the tumor feeding vessel was performed, followed by administration of Cisplatin, Doxorubicin or Adriamycin. 

### 2.4. Radiofrequency Ablation

A triphasic CT scan was performed prior to the procedure to guide the approach to the tumor. An ultrasound-guided percutaneous approach was used for the placement of a 14-gauge needle electrode into the target area. Radiofrequency current was then emitted for 12 to 15 min by a 200 W generator. 

### 2.5. Patient Follow-Up

A CT or MRI scan was obtained from all patients one month following the procedure to document treatment response. Treatment response was assessed using the modified RECIST criteria [16]. Clinical evaluation, surveillance liver scans, and laboratory investigations were subsequently performed every 3–6 months to monitor for progressive disease or recurrence. Repeat treatment was performed for patients with an inadequate response to the initial therapy and for those with recurrence or a progressive disease.

### 2.6. Evaluation of Data

The primary outcome evaluated was overall survival, which was defined as the time between HCC diagnosis and death. Secondary endpoints included treatment response in accordance with the modified RECIST criteria, recurrence, and time to progression (TTP). Recurrence was defined as any new onset lesion or progression of lesions originally considered suspicious or metastasis in patients who had demonstrated a complete response at any time during the follow-up. TTP was defined as the time between primary treatment and the first evidence of radiological progression as defined by the modified RECIST criteria. Outcome measures were evaluated for the whole study population followed by subgroup analysis on tumors less than 3 cm. Adverse effects of treatment were monitored throughout the period of admission and recorded.

### 2.7. Statistical Analysis

All statistical analysis was performed using the Statistical Packages for Social Sciences version 25.0 (SPSS Inc., Chicago, IL, USA). A two-sided *p*-value of <0.05 was considered to be statistically significant. The chi-squared test was used for categorical data comparison and the Mann–Whitney test was used for continuous data. The Kaplan–Meier method with log-rank testing was used for the analysis of survival and time to progression.

## 3. Results

### 3.1. Baseline Charateristics

Between March 1989 and September 2013, 224 patients met the inclusion criteria and were entered in the study. TACE was performed for 168 patients while RFA was performed for 56 patients. Patients were predominantly male and Chinese, with a median age of 65 years at the point of diagnosis. About 86% of patients were cirrhotic with the majority being Child-Pugh A. The main etiology of the underlying liver disease was hepatitis B for both groups. Multiple etiologies of chronic liver disease were noted in 11 patients. No significant differences were noted in the background hepatic function or alpha-fetoprotein levels between both cohorts.

The size of tumor was significantly larger (*p* < 0.001) in the cases treated with TACE compared to RFA. The median size of the primary tumor nodule in the cases treated with TACE was 3.8 cm (interquartile range (IQR) 2.2–6.2 cm) while the median size of the primary nodule in cases treated with RFA was 2.1 cm (IQR 1.5–2.7 cm). The baseline characteristics of all the patients treated with TACE and RFA are shown in Table 1.

There were significant differences between the TACE and RFA subpopulations with regard to ethnicity (“Chinese” and “Others”) and primary diagnosis (“Non-Alcoholic Fatty Liver Disease (NAFLD)”). All differences in sociodemographics and baseline hepatic and clinical factors were resolved by the stratification to a tumor size less than 3 cm, as shown in Table 2.

### 3.2. Survival

Overall survival rates at 1, 3 and 5 years in the RFA and TACE groups were 91.5%, 72.8%, 51.5% and 84.7%, 57.6%, 39.8%, respectively. There was no statistical significance between the two groups (Figure 1, *p* = 0.28). When overall survival was stratified by a size of the HCC of less than 3 cm, the median survival for TACE and RFA groups was 48.0 (IQR 21.0–75.0) and 54.0 (IQR 42.0–67.0) months, respectively, with no significant difference between the groups (Figure 2, *p* = 0.69). The main cause of death was due to hepatic failure from the progression of hepatocellular carcinoma.

### 3.3. Time to Progression

Patients treated with RFA had a significantly longer TTP than those treated with TACE, with a median TTP of 9.0 months (IQR 4.0–19.0) and 13.0 months (IQR 8.0–29.0) (*p* = 0.02), respectively (Figure 3). However, in a subgroup of patients with HCCs less than 3 cm, there was no significant difference in TTP between the TACE (median TTP 13.0 months; IQR 4.0–28.0 months) and RFA groups (median TTP 13.0 months; IQR 9.0–22.0 months) (*p* = 0.55, Figure 4).

### 3.4. Chance of Complete Response

HCC treated with RFA was associated with a significantly higher chance of complete response (CR) compared to TACE (83.9% vs. 32.7%, *p* < 0.001). When stratified by size, RFA-treated HCCs again had a significantly higher CR rate for lesions less than 3 cm compared to TACE (82.2% vs. 55.7%, *p* = 0.004).

### 3.5. Recurrence

Recurrence rates were similar between the TACE- and RFA-treated cases in the overall cohort including both large and small hepatomas (40.3% vs. 58.2%, *p* = 0.21). When stratified to HCCs less than 3 cm, there was a significantly lower recurrence rate in patients that were treated by TACE compared to RFA (39.7% vs. 61.3%, *p* = 0.03).

### 3.6. Number of Treatments

The total number of treatment sessions did not differ significantly (*p* = 0.22) between TACE (median = 2.0, IQR 1.0–3.0) and RFA (median = 2.0, IQR 1.0–3.0). Among hepatomas less than 3 cm, the total number of treatment sessions likewise did not differ significantly (*p* = 0.12), with a median of 2 sessions for TACE (IQR 1.0–2.0) and RFA (IQR 1.0–2.0).

### 3.7. Adverse Events

TACE resulted in an adverse event rate of 37% with 7.4% of TACE patients having a greater number of multiple complications. RFA resulted in single adverse events in 16.1% of patients. Of these, the majority were minor adverse events such as pyrexia, nausea, vomiting, abdominal discomfort and elevated transaminases. For tumors <3 cm, adverse events occurred in 17.7% of TACE patients while RFA patients had adverse events 13.3% of the time. Three episodes of major adverse events were associated with TACE of tumors >3 cm, namely one case of hepatorenal syndrome and two cases tumor rupture. RFA resulted in one major adverse event of pneumothorax in a patient with an HCC <3 cm. Three patients died soon after receiving TACE (1.6%), while no patients died after receiving RFA.

## 4. Discussion

A substantial proportion of HCC patients presenting to tertiary hospitals require loco-regional therapy. Our study shows that treatment with either TACE or RFA for tumors less than 3 cm did not result in a significant difference in overall survival. This is despite a significantly favorable chance of a complete response following RFA therapy.

In accordance with guidelines, patients in our center with larger tumors and a more advanced BCLC stage tend to be treated with chemoembolization while the smaller tumors are primarily managed with RFA. Even so, there exist factors limiting the use of RFA in these patients: central tumors close to the hepatic hilum are at an increased risk of damage to major biliary structures while those peripherally situated adjacent to extrahepatic organs are liable to heat injuries, such as pleural effusion and intestinal perforation [17]. Incomplete ablation may occur in tumors contiguous to large vessels due to tissue cooling caused by increased circulation [18]. Needle-track seeding is a further consideration that has yet to be addressed convincingly [19]. In these cases, patients were treated with TACE. This trend is especially relevant in Asia where TACE has traditionally been favored as the primary anticancer therapy, and hence is even utilized in treating lesions outside the current guidelines [19].

The outcomes at our center were comparable to published studies [20,21]. Our 3-year overall survival rate of 72.8% for RFA and 57.6% for TACE is in line with the pooled 3-year survival rate of 50.8% described in a systematic review by Singal et al. [20]. TACE was used for patients with a larger tumor load, which could explain the difference in overall survival between TACE and RFA. After stratification for smaller tumors less than 3 cm there was resolution of the survival disparity, with no significant difference seen in median survival between patients treated with TACE and RFA.

The examination of secondary clinical outcomes in the subgroup analysis of hepatomas less than 3 cm provides a possible explanation for the comparable survival between RFA and TACE in small tumors. RFA in small tumors has a higher chance of a complete response compared to TACE, though patients who respond to TACE benefit from a lower recurrence. RFA offers a high chance of a complete response in small tumors due to the ability to completely ablate liver tissue and all tumoral residues in the area of the burn. There are, however, risks of tumor seeding, which may increase recurrence [19]. In contrast, the efficacy of TACE depends on arterial supply, and incomplete response may occur if there is more than a single supply and incomplete embolization. This effect is higher in larger and multinodular tumors. There have been concerns that the hypoxia induced by TACE induces vascular endothelial growth factor (VEGF) expression, promoting neovascularization and tumor recurrence in patients incompletely treated by TACE [22]. However, in patients who do respond to TACE, the lower recurrence rates in patients could be due to several reasons: (1) Early recurrence would primarily come from microsatellite and microvascular invasion before treatment [23,24]. Many of these are too small to be identified even with high-resolution imaging and may extend past the safety margin of ablation [25,26]. In such a setting, TACE is able to control these micro-metastases missed by radiofrequency ablation; (2) De novo tumors do arise in cirrhotic livers and may occur in the vicinity of previously treated areas, reflecting a milieu that is favorable for carcinogenesis [24,27,28,29]. The inadvertent spill-over treatment of these high-risk regions with a field change effect downstream of the intended TACE-targeted area may decrease the chance of de novo tumorigenesis and thus explain the longer TTP.

In the era of individualized HCC therapy, there are implications for our results on the treatment of early-stage HCC. The current guidelines advocate for RFA over TACE for the management of HCCs smaller than 3 cm [7,8]. Our study showed that for HCCs less than 3 cm, TACE could lead to a comparable overall survival and TTP compared to RFA. There have been three other retrospective comparative studies of RFA and TACE in small tumors <3 cm or within the Milan criteria, which showed similar overall survival in both groups [12,13,14]. However, the study by Hsu et al. reported a poorer long-term survival in the subgroup undergoing TACE with a total tumor volume <11 cm^3^ [12]. In this study, the patients who underwent TACE had a very high mean AFP of 3175 (ng/mL) compared to 320 in the RFA group. Therefore, the TACE group in the study by Hu et al. may have included tumors with a more aggressive biology.

We acknowledge that limitations exist in this study. All the patients in our study had low AFP, which may suggest favorable biology and impact tumor response. This could also explain the differing findings from Hsu et al. and our results have to be cautiously applied to patients with significantly elevated AFP [12]. Due to its retrospective nature and lack of randomization, our study is unable to provide as strong a conclusion as a randomized controlled trial. Our study is also limited by the small numbers in the subgroup analysis. Even so, a direct comparison between TACE and RFA through a randomized trial has been difficult due to the ethical implications imposed by the present treatment guidelines. We hence hope that our study adds to the body of literature supporting the feasibility of TACE in small HCCs less than 3 cm, especially when other options are unfeasible.

## 5. Conclusions

In summary, we found that in patients with early-stage HCC, both TACE and RFA led to similar overall survival and recurrence rates. This suggests that TACE may be considered as an alternative treatment option in patients unsuitable for surgery and/or RFA.

## Figures and Tables

**Figure 1 biomedicines-10-02361-f001:**
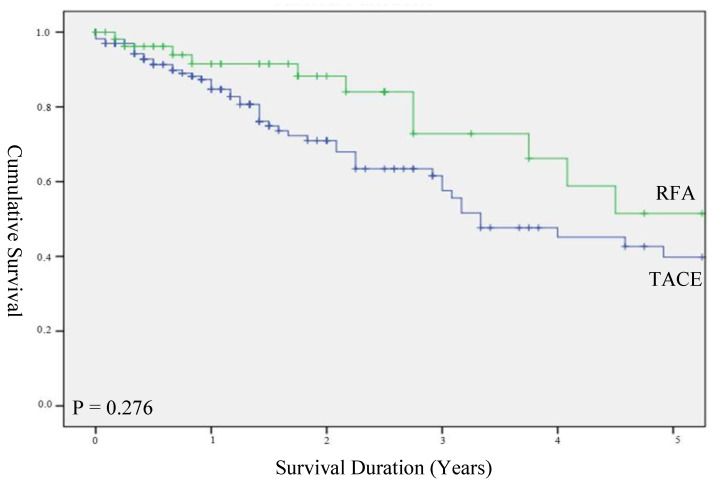
Overall survival in unstratified cohort.

**Figure 2 biomedicines-10-02361-f002:**
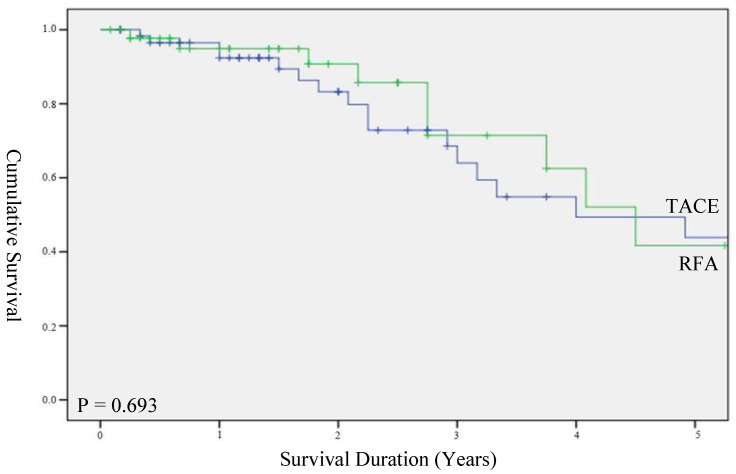
Survival duration stratified for HCC < 3 cm.

**Figure 3 biomedicines-10-02361-f003:**
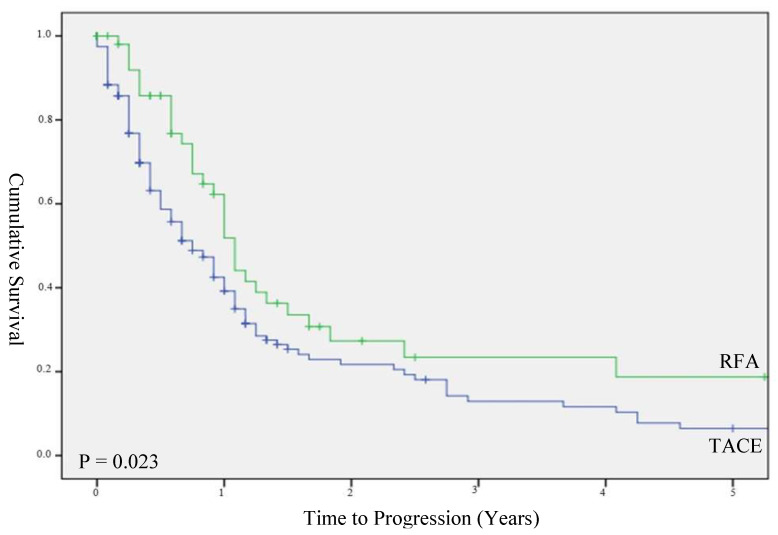
Time to progression in unstratified cohort.

**Figure 4 biomedicines-10-02361-f004:**
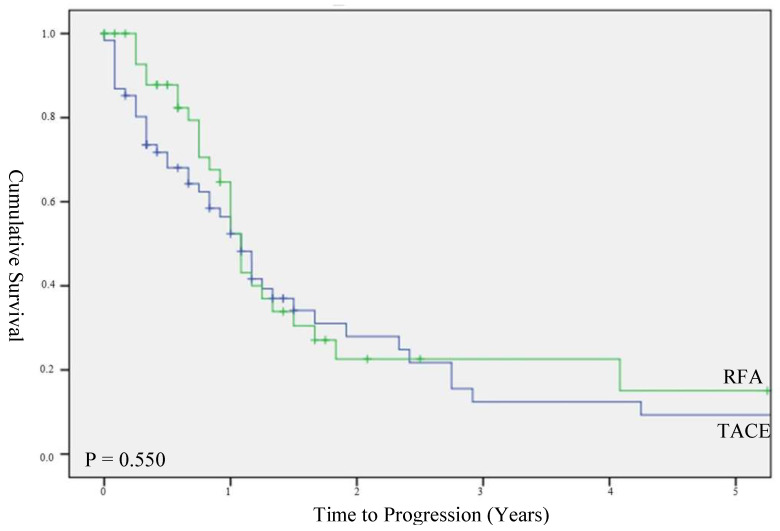
Time to progression stratified for HCC < 3 cm.

**Table 1 biomedicines-10-02361-t001:** Baseline characteristics of all 224 study subjects.

	TACE		RFA		*p* Value
**Number of patients**	168		56		
**Age, median (IQR)**	68	(57–75)	65	(59–70)	0.173
**Gender (%)**					
**Male**	123	(73.2%)	43	(76.8%)	0.597
**Female**	45	(26.8%)	13	(23.2%)	
**Ethnicity (%)**					
**Chinese**	95	(56.5%)	45	(80.4%)	0.012
**Malay**	9	(5.4%)	2	(3.5%)	
**Indian**	6	(3.6%)	0	-	
**Others**	58	(34.5%)	9	(16.1%)	
**Cirrhosis (%)**					
**Yes**	167	(86.8%)	47	(83.9%)	0.564
**No**	22	(13.1%)	8	(14.3%)	
**Unknown**	1	(0.6%)	1	(1.8%)	
**Etiology of underlying liver disease (% of total etiologies)**					
213 cases have a single etiology while 11 cases have multiple etiologies.					
**Hepatitis B**	78	23			0.002
**Hepatitis C**	34	13			
**Alcoholic Cirrhosis**	11	12			
**NAFLD**	12	8			
**Autoimmune Hepatitis**	1	1			
**Wilson’s disease**	2	0			
**Primary Biliary Cirrhosis**	0	2			
**Etiology not known**	35	4			
**Biochemistry**					
**Alpha-fetoprotein (ng/mL)**	18.0	(5.0–122.0)	12.5	(7.0–58.0)	0.460
**Prothrombin Time (s)**	14.0	(13.0–16.0)	15.0	(14.0–16.0)	0.193
**Platelet × 10^9^/L**	128.0	(91.0–211.0)	132.5	(82.5–171.3)	0.339
**Total Bilirubin (umol/L)**	16.0	(10.0–26.25)	17.0	(11.0–33.75)	0.283
**Albumin (g/L)**	36.0	(31.0–40.0)	36.0	(31.0–40.0)	0.870
**Child-Pugh Score (%)**					
**A**	124	(73.8%)	41	(73.2%)	0.930
**B**	44	(26.2%)	15	(26.8%)	
**Tumor Nodularity**					
**Uninodular**	95	(56.5%)	34	(60.7%)	0.748
**Multinodular**	71	(43.5%)	22	(39.3%)	
**Diffuse**	2	(1.2%)	0	-	
**Primary nodule characteristics**					
**Median Size (cm)**	3.8	(2.2–6.2)	2.10	(1.5–2.7)	<0.001

Continuous variables are presented as median (IQR) and categorical variables as n (%).

**Table 2 biomedicines-10-02361-t002:** Baseline characteristics of patients with primary HCC tumor <3 cm.

	TACE		RFA		*p* Value
**Number of patients**	62		45		0.100
**Age**	64	(57–72)	65	(59–70)	0.880
**Gender (%)**					
**Male**	40	(64.5%)	34	(75.6%)	0.222
**Female**	22	(35.5%)	11	(24.4%)	
**Ethnicity (%)**					
**Chinese**	34	(54.8%)	37	(82.2%)	0.018
**Malay**	3	(4.8%)	2	(4.4%)	
**Indian**	3	(4.8%)	0	-	
**Others**	22	(35.5%)	6	(13.3%)	
**Etiology of underlying liver disease (% of total etiologies)** 101 cases had single etiology while 6 cases had multiple etiologies					
**Hepatitis B**	24		19		0.2184
**Hepatitis C**	18		10		
**Alcoholic Cirrhosis**	7		10		
**NAFLD**	6		6		
**Autoimmune Hepatitis**	0		1		
**Wilson’s disease**	1		0		
**Primary Biliary Cirrhosis**	0		2		
**Etiology not known**	6		1		
**Biochemistry**					
**Alpha-fetoprotein (ng/mL)**	16.5	(5.8–99.0)	12.0	(7.0–46.0)	0.781
**Prothrombin Time (s)**	14.0	(13.0–15.0)	15.0	(14.0–15.0)	0.229
**Platelet × 10^9^/L**	108.0	(73.5–160.5)	137.0	(91.0–169.5)	0.100
**Total Bilirubin (umol/L)**	15.0	(11.75–27.0)	15.0	(10.0–27.0)	0.865
**Albumin (g/L)**	37.0	(31.5–40.5)	36.4	(32.5–40.0)	0.597
**Tumor Nodularity**					
**Uninodular**	34	(54.8%)	26	(57.8%)	0.762
**Multinodular**	28	(45.2%)	19	(42.2%)	

Continuous variables are presented as median (IQR) and categorical variables as n (% of total population).

## Data Availability

Kindly contact corresponding author for information requests.

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
