# Peer review of "Comparable Outcomes in Early Hepatocellular Carcinomas Treated with Trans-Arterial Chemoembolization and Radiofrequency Ablation"

_biomedicines, 2022, doi:10.3390/biomedicines10102361_

Round 1
Reviewer 1 Report
The authors present the outcomes from a retrospective analysis of patients with early HCC undergoing RFA vs TACE. A total of 62 TACE patients were compared to 45 RFA patients. Based on their outcomes the authors conclude that in patients with early-stage HCC, both TACE and RFA led to similar overall survival and recurrence rates.
This would have been an interesting study if it had been published 10 years ago. It is somewhat well-written nevertless has some significant issues.
- The authors cite the study of Hsu et al. and quote "Hsu et al. reported a poorer long-term survival in the subgroup undergoing TACE with total tumor volume <11 cm3 [7]. In this study, the patients who underwent TACE had a very high mean AFP of 3,175 (ng/mL) compared to 320 in the RFA group. Therefore, the TACE group in the study by Hu et al. may have included tumors with more aggressive biology." If someone was to read that article more in depth they would see that there was a subgroup analysis of 101 vs 101 PSM matched patients with similar AFP. Moreover they would see that the authors in 2011 showed a similar outcome to this study for patients with <3cm tumors making the current study's outcomes not really novel.
- Patient baseline characteristics are poorly presented
- Kim et al. paper is not mentioned and not cited.
- A significant number of recent studies present good outcomes of TACE+RFA, does this not make the outcomes and modalities presented a bit out of date in clinical practice?
- The number of patients included is really small and in my view proposing "This challenges the traditional belief that RFA is the first-line option for small HCC less than 3cm in patients for whom surgery are not feasible or preferred and TACE may be considered as an alternative treatment for this group of patients, especially when technical or patient factors limit the use of RFA." is really bold and should be revised.
- All patients had really low AFP thus as the authors cite that ALL included patients may have had favorable biology.
- Were all the patients confirmed cirrhotic?
- The limitations section is rather poor
Author Response
The authors present the outcomes from a retrospective analysis of patients with early HCC undergoing RFA vs TACE. A total of 62 TACE patients were compared to 45 RFA patients. Based on their outcomes the authors conclude that in patients with early-stage HCC, both TACE and RFA led to similar overall survival and recurrence rates.
This would have been an interesting study if it had been published 10 years ago. It is somewhat well-written nevertless has some significant issues.
- The authors cite the study of Hsu et al. and quote "Hsu et al. reported a poorer long-term survival in the subgroup undergoing TACE with total tumor volume <11 cm3 [7]. In this study, the patients who underwent TACE had a very high mean AFP of 3,175 (ng/mL) compared to 320 in the RFA group. Therefore, the TACE group in the study by Hu et al. may have included tumors with more aggressive biology." If someone was to read that article more in depth they would see that there was a subgroup analysis of 101 vs 101 PSM matched patients with similar AFP. Moreover they would see that the authors in 2011 showed a similar outcome to this study for patients with <3cm tumors making the current study's outcomes not really novel.
We thank you for your kind feedback. We do recognize that the study may be less novel, however we aim to provide real world data from South East Asia to inform clinical practice.
- Patient baseline characteristics are poorly presented
We thank you for your advice and have made revisions to the presentation of baseline characteristics. Further characteristics are presented in table 1.
- Kim et al. paper is not mentioned and not cited.
Thank you for the guidance. We have included Kim et. al. paper in lines 22 to 25 and included the citation (8,9)
- A significant number of recent studies present good outcomes of TACE+RFA, does this not make the outcomes and modalities presented a bit out of date in clinical practice?
We do agree that clinical practice has expanded to encompass combined TACE+RFA. However, there are cases where RFA may not be technically feasible and hence our study aims to offer TACE as a suitable alternative if other modalities are unsuitable. Our study provides real world data from South East Asia and helps to inform practice. There has been a recent publication in 2021 comparing outcomes between single DEB-TACE and RFA for single small HCC 3cm or less with similar findings (https://doi.org/10.17998/jlc.2021.05.20)
- The number of patients included is really small and in my view proposing "This challenges the traditional belief that RFA is the first-line option for small HCC less than 3cm in patients for whom surgery are not feasible or preferred and TACE may be considered as an alternative treatment for this group of patients, especially when technical or patient factors limit the use of RFA." is really bold and should be revised.
We concur with your suggestion and have suitably revised the statement to “This suggests that TACE may be considered as an alternative treatment option in patients unsuitable for surgery and/or RFA”
- All patients had really low AFP thus as the authors cite that ALL included patients may have had favorable biology.
We do agree that the overall low AFP may have impacted the findings and have expanded the limitations section with this astute judgement.
- Were all the patients confirmed cirrhotic?
86.8% of TACE patients and 83.9% of RFA patients were cirrhotic with no significant difference between both subgroups (P 0.564). We have added this information into table 1 and in the results section.
- The limitations section is rather poor
We thank you for your feedback and have expanded the contents of the limitations section to encompass small sample size, retrospective study nature and lack of randomization in addition to the overall low AFP of the study population
Reviewer 2 Report
The authors compared the outcomes of TACE and RFA in patients with HCC less than 3 cm. They showed that there was no significant difference in overall survival, time-to-progression and number of treatment sessions required. The topic of manuscript is interesting, but I have several considerable concerns.
Major comments
#1. Although the authors intended to compare the outcomes of TACE and RFA in patients with HCC less than 3 cm, they included 106 patients for TACE and 11 patients for RFA with HCC of 3 cm or larger.
#2. Abstract. They described that recurrence rates are similar between RFA and TACE groups after propensity score matching analysis. They did not show the comparison of recurrence rate in the results.
#3. The baseline characteristics of the patients used for propensity score matching analysis should be presented.
#4. The numbers of patients with HCC less than 3 cm (62 and 45) and propensity score matching analysis (56 and 56) are too small for analysis of difference of treatment outcomes.
#5. P5, lines 5 and 6. For propensity score matching analysis, the authors included 11 patients for RFA with HCC of 3 cm or larger. This is inappropriate, since they discuss the outcomes of TACE and RFA in patients with HCC less than 3 cm.
#6. Results. They did not show the comparison of time to progression, complete response rate, recurrence rate, and number of treatments of propensity score matching analysis.
#7. Adverse events. They should describe the adverse events which developed in all the patients, the patients with HCC less than 3 cm, and the patients for propensity score matching analysis, separately.
#8. P8, lines 35-37. Are the percentages of 91.5%, 84.7%, 41.2%, and 39.6% derived from ref.17 or from the results of the present study?
#9. P9, lines 32-33. On the contrary to this description, they showed a significantly lower recurrence rate in HCC less than 3cm that were treated by TACE than RFA (39.7% vs 61.3%, p = 0.03).
Author Response
The authors compared the outcomes of TACE and RFA in patients with HCC less than 3 cm. They showed that there was no significant difference in overall survival, time-to-progression and number of treatment sessions required. The topic of manuscript is interesting, but I have several considerable concerns.
Major comments
#1. Although the authors intended to compare the outcomes of TACE and RFA in patients with HCC less than 3 cm, they included 106 patients for TACE and 11 patients for RFA with HCC of 3 cm or larger.
Our study aims to provide our center experience and outcomes for A) patients treated with TACE and RFA followed by B) comparison in the subgroup of patients with HCC of less than 3cm. The paper hence encompassed findings from the entire cohort regardless of size followed by analysis of the subgroup of less than 3cm. We note your kind feedback on the potential for confusion and have amended the wording to provide greater resolution in the subsection (evaluation of data) in lines 79-80
#2. Abstract. They described that recurrence rates are similar between RFA and TACE groups after propensity score matching analysis. They did not show the comparison of recurrence rate in the results.
We thank you for your feedback. We have amended the manuscript to exclude the propensity matched analysis as focus of the study is on comparison of the entire cohort followed by the subgroup of HCC <3cm.
#3. The baseline characteristics of the patients used for propensity score matching analysis should be presented.
We thank you for your feedback. We have amended the manuscript and excluded the propensity matched analysis as the focus of this study is on comparison of the entire cohort followed by the subgroup of HCC <3cm.
#4. The numbers of patients with HCC less than 3 cm (62 and 45) and propensity score matching analysis (56 and 56) are too small for analysis of difference of treatment outcomes.
We too concur on this limitation. We recognize the limitation of the small sample size and have expanded the limitation to encompass this.
#5. P5, lines 5 and 6 For propensity score matching analysis, the authors included 11 patients for RFA with HCC of 3 cm or larger. This is inappropriate, since they discuss the outcomes of TACE and RFA in patients with HCC less than 3 cm.
We thank you for your feedback. We have amended the manuscript and excluded the propensity matched analysis as the focus of this study is on comparison of the entire cohort followed by the subgroup of HCC <3cm.
#6. Results. They did not show the comparison of time to progression, complete response rate, recurrence rate, and number of treatments of propensity score matching analysis.
We thank you for your feedback. We have amended the manuscript and excluded the propensity matched analysis as the focus of this study is on comparison of the entire cohort followed by the subgroup of HCC <3cm.
#7. Adverse events. They should describe the adverse events which developed in all the patients, the patients with HCC less than 3 cm, and the patients for propensity score matching analysis, separately.
We thank you for the advice and have amended the adverse event section to include data on HCC < 3cm. Have excluded the propensity score matched analysis from this section as well
#8. P8, lines 35-37. Are the percentages of 91.5%, 84.7%, 41.2%, and 39.6% derived from ref.17 or from the results of the present study?
We thank you for your clarification. These percentages are derived from the results of our present study. These are comparable to published studies summarized by Singal et.al.
#9. P9, lines 32-33. On the contrary to this description, they showed a significantly lower recurrence rate in HCC less than 3cm that were treated by TACE than RFA (39.7% vs 61.3%, p = 0.03).
We thank you for the clarification. We aimed to present the recurrence rate in the overall cohort followed by the subgroup analysis in HCC <3cm. There was no significant difference in recurrence rate in the overall cohort but a significantly lower recurrence rate in patients with HCC < 3cm that were treated by TACE compared to RFA. We have amended the section to clearly reflect this.
Round 2
Reviewer 1 Report
In their revised version the authors adressed the majority of my remarks.
Author Response
We thank you for your kind review
Reviewer 2 Report
In my former review, I indicated the inappropriate presentation of the propensity score matching analysis. The authors excluded the description of the propensity score matching analysis. I still have three comments.
#1. Abstract. Although the authors say that they excluded the propensity score matching analysis in the revised manuscript, there is still the description of the propensity score matching analysis.
#2. P14, lines 198-200. The percentages of 91.5%, 84.7%, 41.2%, and 39.6% differ from 91.5%, 84.7%, 51.5%, and 39.8% in P11, lines 129-130. In addition, they should present the percentages from the study by Singal et al.
#3. They should mainly discuss the data of the patients with HCC less than 3cm, which is more reliable than the data of the entire cohort including 106 patients for TACE and 11 patients for RFA with HCC of 3cm or larger.
Author Response
In my former review, I indicated the inappropriate presentation of the propensity score matching analysis. The authors excluded the description of the propensity score matching analysis. I still have three comments.
#1. Abstract. Although the authors say that they excluded the propensity score matching analysis in the revised manuscript, there is still the description of the propensity score matching analysis.
We thank you for your kind review and have revised the manuscript to ensure exclusion of the propensity matching analysis.
#2. P14, lines 198-200. The percentages of 91.5%, 84.7%, 41.2%, and 39.6% differ from 91.5%, 84.7%, 51.5%, and 39.8% in P11, lines 129-130. In addition, they should present the percentages from the study by Singal et al.
We thank you for identification of this discrepancy. The 5-year survival of 51.5% and 39.8% is the accurate set of figures. However, Singal et primarily presented 3-year survival data. We hence have revised the discussion segment accordingly to reflect 3-year survival data. This now reads: “Outcomes at our centre were comparable to published studies. Our 3-year overall survival rate of 72.8% for RFA and 57.6% for TACE is in line with pooled 3-year survival rate of 50.8% described in a systematic review by Singal et. al”
#3. They should mainly discuss the data of the patients with HCC less than 3cm, which is more reliable than the data of the entire cohort including 106 patients for TACE and 11 patients for RFA with HCC of 3cm or larger.
We thank you for the advice and have amended the manuscript, in particular the discussion segment to streamline focus on the subset of HCC <3cm.